# The Role of Ultrasound in the Preparation of Zein Nanoparticles/Flaxseed Gum Complexes for the Stabilization of Pickering Emulsion

**DOI:** 10.3390/foods10091990

**Published:** 2021-08-25

**Authors:** Yinghao Li, Ge Xu, Weiwei Li, Lishuang Lv, Qiuting Zhang

**Affiliations:** Department of Food Science and Technology, School of Food Science and Pharmaceutical Engineering, Nanjing Normal University, Nanjing 210023, China; yinghaoli96@163.com (Y.L.); xuge813@163.com (G.X.); liweiwei@njnu.edu.cn (W.L.); lishuanglv@126.com (L.L.)

**Keywords:** zein, flaxseed gum, ultrasound, complex particle, Pickering emulsion

## Abstract

Ultrasound is one of the most commonly used methods to prepare Pickering emulsions. In the study, zein nanoparticles-flaxseed gum (ZNP-FSG) complexes were fabricated through various preparation routes. Firstly, the ZNP-FSG complexes were prepared either through direct homogenization/ultrasonication of the zein and flaxseed gum mixture or through pretreatment of zein and/or flaxseed gum solutions by ultrasonication before homogenization. The Pickering emulsions were then produced with the various ZNP-FSG complexes prepared. ZNP-FSG complexes and the final emulsions were then characterized. We found that the complex prepared by ultrasonication of zein as pretreatment followed by homogenization of the ZNP with FSG ((ZNP_U_-FSG)H) exhibited the smallest turbidity, highest absolute potential value, relatively small particle size, and formed the most stable complex particles. Meanwhile, complex prepared through direct ultrasonication plus homogenization on the mixture ((ZNP-FSG)HU) showed significantly decreased emulsifying properties and stability. Compared with the complex without ultrasonic treatment, the complex and emulsion, which prepared by ultrasonicated FSG were extremely unstable, and the phase separation phenomenon of the emulsion was observed 30 min after preparation. The above conclusions are also in line with the findings obtained from the properties of the corresponding emulsions, such as the droplets size, microstructure, freeze-thaw stability, and storage stability. It is, therefore, clear that to produce stable Pickering emulsion, ultrasonication should be avoided to apply together at the end of ZNP-FGS complex preparation. It is worth noticing that the emulsions prepared by complex with ultrasonicated zein (ZNP_U_-FSG)H are smaller, distributed more uniformly, and are able to encapsulate oil droplets well. It was found that the emulsions prepared with ZNP_U_-FSG remained stable without serum phase for 14 days and exhibited improved stability at low-temperature storage. The current study will provide guidance for the preparation of protein–polysaccharide complexes and Pickering emulsions for future work.

## 1. Introduction

The emulsions stabilized by aqueous solid particles are termed Pickering emulsions, which is different from those stabilized by small-molecule surface emulsifiers. Solutions of many organic ingredients, such as protein, glue, flour, milk, starch, saponin, could act as emulsifiers for Pickering emulsion [1]. Pickering emulsions prepared using food-grade colloidal particles can be designed to have high physical stability and overcome the problems of low biocompatibility [2] and environmental pollution [3] caused by traditional synthetic surfactants. With the increasing awareness and pursuit of health and safety from consumers, formulating food with natural materials has been increasingly expected from academia to the food industry. Protein-based colloidal particles are particularly suitable since they are widely available, inexpensive, and show great nutritional benefits [4]. Recently, there are many pieces of research on Pickering emulsions using different colloidal particles, such as starch [5], polysaccharides [6], protein [7], and phospholipids, to form combinations to pursue the emulsification effect closer to that of synthetic chemical surfactants [8]. However, research focusing on the influence of the emulsion preparation process on emulsion stability is still far from enough.

Many techniques have been applied for emulsion preparation, such as high shear mixers (homogenization), ultrasound [9] micro-fluidization, and high-pressure homogenization [10]. Homogenization and ultrasound are the two most commonly used methods to prepare Pickering emulsions. Ultrasound has the advantages of high efficiency, good reproducibility, low cost, simple operation, no pollution, and being able to maintain the internal nutrients of food, which makes the ultrasonic technology widely used. Many reports show ultrasonic treatment could improve the emulsification property of the emulsion stabilized by soy protein isolate [11], chitosan [12], and whey protein-pectin complexes [13]. Moreover, high-intensity ultrasound is more energy-efficient than high-pressure homogenization and microfluidizer [14]. However, the effects of ultrasound on the stability of emulsion were rarely characterized in detail, e.g., effects of ultrasonic parameters, substrates, different processing procedures. In addition, there are few studies on the interaction between different emulsion preparation methods.

In our previous study, we found ultrasonic pretreatment could improve the stability of zein stabilized Pickering emulsion. But if ultrasound is directly applied to zein nanoparticles (ZNP), which is prepared by anti-solvent precipitation method, the stability of ZNP would not have been improved after ultrasonic treatment [15]. Julius W. J. de Folter et al. demonstrated that ZNP system synthesized using anti-solvent precipitation procedure is very stable, specifically at a pH of 3–4. However, Pickering emulsions stabilized by ZNP failed to form stable emulsions due to extreme environmental stresses (pH, heating, and salt) [16]. To improve the stability of the ZNP emulsion, forming a complex with other hydrophilic molecules, such as hydrophilic protein, polysaccharides, or polyphenols, is a common method. Among them, polysaccharides, including sodium alginate, low-acyl gellan gum, gum arabic, chitosan, and carboxymethyl dextrin have been reported to be effective [17,18,19,20]. Flaxseed gum (FSG), with good gelling, emulsifying, and rheological properties, was used to form complexes with ZNP [21,22,23,24]. Although our previous research showed that flaxseed gum could improve the stability of ZNP emulsion, the influences of different preparation methods and different combinations of ultrasound and homogenization on emulsion have not been studied. Therefore, in the current study, we studied the preparation of zein nanoparticles-flaxseed gum (ZNP-FSG) complexes according to different procedures, including either applying ultrasonication and/or homogenization of the mixed solution without any previous treatment or pretreatment using ultrasonic of one or both ingredients. Both the ZNP-FSG complexes and the final emulsions were characterized. The aim is to devise the preparation route of ZNP-FSG complexes that are best suited for the stabilization of Pickering emulsions [25].

## 2. Materials and Methods

### 2.1. Materials

Zein was purchased from Adamas Reagent Co., Ltd. (Shanghai, China). Flaxseed gum was provided by Yuanye Biotechnology Co., Ltd. (Shanghai, China). Corn oil was obtained from a local supermarket (Nanjing, China). Other reagents were analytical grade.

### 2.2. Preparation and Pretreatment of Samples 

In order to analyze the role of ultrasound on Pickering emulsion preparation, different combinations of ultrasound and homogenization and different ultrasound ingredients were compared, 1. zein, 2. FSG, 3. zein-FSG mixtures. 

#### 2.2.1. Preparation Individual Solution 

Zein stock solution, which was zein dissolved in 70% (*v*/*v*) ethanol at 25 mg/mL followed by continuous shaking for 1 h at 25 °C prior to ultrasound treatment. 

FSG stock solution was completely dissolved in water at 5 mg/mL by magnetic stirring at 95 °C for 1 h, the pH was adjusted to 4.0 with 0.1 M HCl or NaOH solutions and measured by a pH meter. 

#### 2.2.2. Preparation of Zein Nanoparticles (ZNP)

Zein nanoparticles (ZNP) were prepared by anti-solvent precipitation method according to Dai, L. [26]. Briefly, 20 mL of zein stock solution was added dropwise into 60 mL deionized water in an ultrasonic water bath (KH-300DE, Kunshan Hechuang Ultrasonic Instrument Co., Ltd., Kunshan, Jiangsu, China). After magnetic stirring for 30 min, the remained ethanol in the particle dispersions was removed by rotary evaporation (IKA RV 8, IKA Works Guangzhou, China). The final concentration of ZNP was 10 mg/mL. Finally, the pH value of ZNP dispersions was adjusted to 4.0 with 0.1 M HCl or NaOH solutions using a pH meter and stored at 4 °C until use. The nanoparticles are made from natural or ultrasonic zein separately. 

#### 2.2.3. Ultrasound Treatment 

The sample was subjected to ultrasonic treatment with a frequency of 20 kHz sonicator (Model JY92-IIN, NingBo Xinzhi Technology Co., Ningbo, China), which was equipped with a titanium probe (6 mm of diameter) in an ice bath. 

A volume of 20 mL of zein solution (25 mg/mL, *w*/*v*) and 20 mL of FSG (5 mg/mL, *w/v*) was treated, respectively, by ultrasound at 52.95 W·cm^−2^ for 10 min, with pulsed ratios was 5 s on-time/5 s off-time (U). Zein or FSG solution ultrasonicated were termed ZNP_U_ or FSG_U_, respectively. 

### 2.3. Preparation of Complex Particles

The complex particles were prepared using different ZNP and FSG by combination with homogenization and/or ultrasonication, and detailed procedures are shown in Figure 1 and Table 1. The differently treated samples were stored at 4 °C until use. 

A volume of 20 mL of ZNP-FSG complexes was also treated by ultrasound at 52.95 W·cm^−2^ for 10 min, with pulsed ratios of 5 s on-time/5 s off-time (U). ZNP-FSG solution ultrasonicated was termed (ZNP-FSG)U. 

Samples needed to be treated by a high-speed dispersion homogenizer (FJ200-SH, Shanghai specimen model factory, Shanghai, China), the homogenizing conditions were fixed at 15,000 rpm for 3 min, marked as H.

### 2.4. Preparation of Pickering Emulsions

Pickering emulsions were prepared as described by Yang et al. [27] with slight modifications. The emulsions were prepared by mixing 10 mL of complex and 10 mL corn oil and then homogenized by a high-speed homogenization at 18,000 rpm for 2 min. The resultant emulsions were stored at 4 °C for further characterization.

### 2.5. Characterization of Complexes

#### 2.5.1. Particle Size and Zeta (ζ)-Potential of Complexes

According to the method of Lv et al. [28], the particle size of the complexes was analyzed by dynamic light scattering using a Zetasizer (Nano-ZS90, Malvern Instruments Co., Ltd., Worcestershire, UK) at a fixed detector angle of 90°. Samples were diluted 50 times by the distilled water at pH 4 adjusted by 0.1 M HCl or NaOH solutions. They were equilibrated for 60 s before detection. The instrument reports the mean particle diameter (z-average) and the polydispersity index (PDI) ranging from 0 to 1. PDI measures the broadness of size distribution, and it provides information on the deviation from mean particle size [29].

Similar to particle size measurement, the ζ-potentials (mV) of the complexes were determined by measuring the electrophoretic mobility of each solution using a Malvern Zetasizer at 25 °C in triplicate. 

#### 2.5.2. Turbidity Measurement

Turbidity was determined according to Hosseini [30]. The transmittance of samples in distilled water at pH 4.0 with 0.1 M HCl or NaOH solutions was measured at 600 nm by a UV-Vis spectrophotometer (TU-1810PC, Beijing Purkinje General Instrument Co., Ltd., Beijing, China). A turbidity of 0% corresponds to a totally clear solution (transmittance in comparison to the blank is 100%).

#### 2.5.3. Salt Ionic Stability Measurement

The effect of ionic strength on the formation of complexes was determined as described by Lv et al. [28]. The absorbance value of samples with different salt ion concentrations (50, 100, 200, 300, 400, 500, and 1000 mmol/L) was measured at 600 nm to study the stability changes.

#### 2.5.4. Surface Hydrophobicity (H_0_)

Surface hydrophobicity of the complex particles was conducted on the basis of the description of Kato et al. [31]. The samples were diluted into a series of concentrations (0.2, 0.4, 0.6, 0.8, and 1 mg/mL) with deionized water with pH adjusted to 4.0 using 0.1 M HCl or NaOH solutions. Then, 50 μL of 1-anilino-8-naphthalene-sulfonate (ANS) solution (8 mM in deionized water of pH 4.0) was mixed with 5 mL of the diluted sample incubated in the darkroom (25 °C) for 15 min. The relative fluorescence intensity of the sample was determined by a fluorescence spectrophotometer (F-380, Tianjin GangDong scientific and technology Co., Ltd., China) at 390 nm (excitation wavelength, slit width 5 nm) and 468 nm (emission wavelength, slit width 5 nm). The initial slope of fluorescence intensity versus protein concentration (%, *w/v*) was calculated by linear regression analysis and used as an index of H_0_.

#### 2.5.5. Determination of Emulsifying Properties

The determination of emulsifying property of complex particles was carried out according to the method of Li et al. [32], slightly changed. A total of 30 mL of samples were homogenized at room temperature at 20,000 r/min for 1 min after adding 10 mL of oil. After homogenization, 50 μL of the sample was taken from the bottom immediately and added into 5 mL 0.1% sodium dodecyl sulfate (SDS). The absorbance of the diluted solution was measured at 500 nm, and 0.1% SDS was used as blank. The absorbance measured immediately (A_0_) and standing for 10 min (A_10_) after emulsion formation were used to calculate the emulsifying activity index (EAI) and emulsion stability index (ESI) as follows.
(1)$$\mathrm{EAI}=2\;\times \;2.303\;\times \frac{A_{0}\;\times \;\mathrm{DF}}{C\;\times \;\varphi \;\times \;\theta \;\times \;10000}$$
(2)$$\mathrm{ESI}=\frac{A_{0}}{A_{0}-A_{10}}\;\times \;∆t$$
where A_0_ is the absorbance at 0 min; DF is the dilution factor (100); θ is the proportion of the oil phase (0.25); φ is the optical path of the cuvette (1 cm); C is the concentration of complex particles (g/mL); A_10_ is the absorbance at 10 min; ∆t is 10 (min).

### 2.6. Determination of Stability 

#### 2.6.1. Storage Stability of Pickering Emulsions

The stability against the creaming of the emulsions was analyzed via creaming index (CI). The method of Ahmed et al. [11] with slight modifications was used to assess the storage stability of Pickering emulsions. Fresh emulsions (15 mL) were kept in 25 mL-antifreeze glass bottles, then stored at 4 °C for 14 days. And samples were withdrawn for analysis of emulsion properties at regular intervals. The results were calculated using the following equation:(3)$$\mathrm{CI}\;(\%)=\frac{H_{c}}{H_{0}}\;\times \;100$$
where H_c_ represents the height of the serum phase, and H_0_ represents the total height of emulsion.

#### 2.6.2. Freeze-Thaw Stability of Pickering Emulsions

The freeze-thaw stability of Pickering emulsions was measured by the previous method of Zhu [33]. A total of 15 mL of the emulsion was shifted into freeze-resistant glass bottles and then placed in a freezer at −18 °C for 22 h. Then the emulsions were naturally defrosted at room temperature for 2 h. The process was repeated thrice, the creaming index and oiling off index of the emulsions were recorded after each cycle.

### 2.7. Droplets Size of Pickering Emulsions

Droplet size distributions of emulsions were measured by a Mastersizer 3000 (Malvern Instruments Co., Ltd., Worcestershire, UK). The Mie theory was applied by considering the following optical properties for corn oil droplets: a refractive index of 1.46 and absorption of 0.001, and for the deionized water, a refractive index of 1.33. Add the sample drop by drop to the deionized water with automatic mixing until an obscuration rate of between 10% and 20% [34]. Prior to the measurements, the samples were thermostated at 25 °C for 5 min and all measurements were carried out over at least 3 times. All emulsions were analyzed 24 h after the emulsification process. The volume-weighted mean diameter (D_4,3_) was reported.

### 2.8. Microstructure Analysis

The size, shape, and aggregate state of emulsion droplets were quantified by the optical microscopic method described by Yang et al. [27] with slight modifications. Approximately 20 μL of emulsions were deposited on the microscope slide and covered with a coverslip. The morphology of the Pickering emulsion droplet was observed using an inverted optical microscope (CKX41, Olympus, Tokyo, Japan) at 200× magnification. 

Confocal Laser Scanning Microscopy (CLSM) were determined by using a Nikon Ti-E-A1R confocal microscope (Nikon Instruments Inc, Tokyo, Japan) to observe the microstructure of the emulsions. The CLSM was operated in the fluorescence mode. Nile red was used to stain the oil phase. All of the stain solutions were prepared at room temperature and stored in a dark place. Approximately 1.0 mL of the samples were completely mixed with a 10 μL aliquot of the Nile red (0.1%, *w*/*v*). The mixtures were equilibrated for 30 min before observation. Approximately 20 μL of the stained samples was placed on slide and carefully covered with a coverslip, ensuring that there was no trapped air gap or bubbles between the mixture and the cover slip. The images were collected using 488 nm excitation wavelengths for Nile red. Each image contained 1024 × 1024 pixels.

### 2.9. Statistical Analysis

All experiments were performed in triplicate, and the value was expressed as mean value ± standard deviation. The statistical differences among means were evaluated using a one-way analysis of variance and Duncan’s test (*p* < 0.05).

## 3. Results and Discussion

### 3.1. Effects of Ultrasound Process on the Preparation of ZNP-FSG Complexes

#### 3.1.1. Emulsifying Properties

The emulsifying activity index (EAI) refers to the ability of proteins absorbed into the oil/water interface. The emulsifying stability index (ESI) is an indicator of the ability of a protein to maintain emulsion stability [35]. The EAI and ESI of the complexes subjected to various ultrasound and homogenization treatments are presented in Figure 2. The results showed that the EAI and ESI of untreated control sample ZNP were 6.09 ± 0.62 m^2^/g and 12.64 ± 0.43 min, respectively. In comparison with the ZNP, the EAI and ESI of homogenization treated complex (ZNP-FSG)H were significantly increased, up to 10.47 ± 1.20 m^2^/g and 42.47 ± 4.10 min, respectively. Interestingly, no significant difference was observed between the EAI and ESI of (ZNP-FSG)H and (ZNP_U_-FSG)H, indicating the emulsification property of the homogenization-treated complex was independent from the ultrasonic treatment of zein. Moreover, (ZNP-FSG)H and (ZNP_U_-FSG)H exhibited higher EAI and ESI than that of the (ZNP-FSG_U_)H, implying that pretreatment of FSG with ultrasonication was not favorable to improve the emulsifying properties of the complex. More importantly, we found that the (ZNP-FSG)HU and (ZNP-FSG)U complex showed significantly lower EAI and ESI than that of (ZNP-FSG)H. This result indicates that further treatment with ultrasonication could dramatically destroy the emulsifying capability of the complex. In fact, the (ZNP-FSG)H and (ZNP_U_-FSG)H complex had the highest EAI and ESI among all the samples in the current study. Finally, there was no obvious difference among the EAI and ESI between ZNP, (ZNP-FSG)HU, and (ZNP-FSG)U. That is to say, adding FSG did not necessarily improve the emulsification property of ZNP, which also depends on the pretreatment and emulsifying process. The improved emulsifying properties of ZNP-FSG complexes may be correlated with the solubility, structural changes of zein, electrostatic interaction, and hydrophobic interaction between ZNP and FSG [32,36,37].

#### 3.1.2. Turbidity Changes 

Dispersion turbidity indicates the number and size of dispersed particles to some content [38], which reflects the dispersion state and aggregation degree. Figure 3 illustrates the turbidity of ZNP and ZNP-FSG complexes. Turbidity of most complexes decreased after ultrasound pretreatments, except (ZNP-FSG)HU and (ZNP-FSG)U, which increased. It may be because the stable balances of original complexes were destroyed by extra ultrasound. Contrary to our results, Ma et al. [37] found the ultrasound treatment significantly decreased the turbidity of the soy protein isolate (SPI)—citrus pectin (CP) complex. The type of polysaccharide is probably one of the reasons for the different phenomenon. The turbidity of (ZNP_U_-FSG) is lowest, and the turbidity of (ZNP-FSG)H is lower than that of (ZNP-FSG_U_)H. From this, we hypothesized that the particle size changes of proteins induced by ultrasonic treatment decreased the turbidity of the solution most [39], which was related to the smaller protein particle size obtained by ultrasonic treatment. (ZNP-FSG_U_)H might have had higher turbidity because of the degradation of FSG [36].

#### 3.1.3. Salt Ionic Stability 

The stability of the complexes at different ionic strengths (50 mM, 100 mM, 200 mM, 300 mM, 400 mM, 500 mM, and 1000 mM) was evaluated by visual observation and turbidity in Figure 4. In Figure 4a, ZNP started to form flocculation and precipitation at a 50 mM NaCl due to electrostatic screening effects [40]. The flocculation extent kept increasing with increased salt concentration. Different from ZNP, it can be seen from the picture that the stability of the ZNP-FSG complex at different salt ion concentrations improvement was attributed to the addition of FSG, even though differences were noticed among complexes with different treatments. The result indicates that FSG is effective in protecting ZNP from salt-induced aggregation, which is probably due to the weakened electrostatic screening effects of ZNP brought by the electrostatic interaction between ZNP and FSG [40]. However, the tolerance of (ZNP-FSG)HU to salt was relatively poor, and flocculation can be clearly observed, as shown in Figure 4a. 

The turbidity results, as shown from Figure 4b, is in agreement with the visual observation. The turbidity of (ZNP_U_-FSG)H and (ZNP-FSG)H decreased as the NaCl concentration increased. The turbidity of (ZNP-FSG_U_)H, (ZNP-FSG)U, and (ZNP-FSG)HU did not show a great relationship with the increasing salt ion concentration but fluctuated to different degrees. As we can see, the turbidity of (ZNP-FSG)HU and (ZNP-FSG)U showed an obvious upward trend at the salt ion concentration of 1000 mM. This is probably caused by the decreased interaction between ZNP and FSG, which resulted from the electrostatic shielding effect at high ion concentrations. At all the tested ion concentrations, (ZNP_U_-FSG)H complex showed lower turbidity than the others, which indicated that (ZNP_U_-FSG)H had the best tolerance of salt. Overall, the complexes with different treatments had a good stability at the salt ion concentration of 1000 mM except for (ZNP-FSG)HU. Increases of (ZNP-FSG)U and (ZNP-FSG)HU in turbidity illustrate the continued aggregation of soluble complexes into insoluble protein/polysaccharide complexes due to charge neutralization.

#### 3.1.4. Surface Hydrophobicity

Surface hydrophobicity (H_0_) is closely related to the emulsifying properties, stability, and function of the complex/particles [41], and a balanced surface hydrophobicity has a beneficial effect on surface activity and interfacial performance in emulsions. The surface hydrophobicity of the complexes is shown in Figure 5. We concluded that the H_0_ of ZNP could be increased by ultrasonic pretreatment on zein. Ultrasonic cavitation can effectively expose the buried hydrophobic regions of zein from the interior of molecules to the surface [42]. Similar results were obtained by Gao et al. [14], investigating the effect of ultrasound on whey isolated protein. The H_0_ of (ZNP-FSG)H complexes significantly decreased after adding FSG, which was ascribed to FSG blocking the binding sites between the fluorescent probe and hydrophobic groups of protein. Wang et.al [43] also reported that the hydrophobicity of chitosan-zein complexes decreased due to the addition of chitosan. The surface hydrophobicity of (ZNP_U_-FSG)H was higher than that of (ZNP-FSG)H, which indicated that the increase in H_0_ of ZNP after ultrasound improved the H_0_ of (ZNP_U_-FSG)H. The H_0_ value of (ZNP-FSG)HU and (ZNP-FSG)U showed an extreme decrease after sonication, which was independent of whether it had been homogenized or not. This was probably due to the contact between the ultrasound and protein increasing when FSG separates from ZNP after sonication, and hydrophobic groups were reburied. A similar result was reported in the effects of dynamic high-pressure microfluidization treatment on the functional and structural properties of potato protein isolate and its complex with chitosan by Hu et al. [44]. The changing trend of surface hydrophobicity is consistent with that of emulsification capability. Previous research [45] has shown that the adsorption rate was shown to scale with the relative hydrophobicity, which is considered to be helpful to improve emulsifying properties.

#### 3.1.5. Particle Size

In order to study the effects of the pretreatment methods and operating conditions on different samples, the particle size of complexes was characterized, as shown in Figure 6a. Ultrasound pretreatment was able to significantly decrease the particle size of ZNP from 226.1 nm to 185.4 nm, which was possibly related to the structural changes of protein upon ultrasonication [46]. The particle size of FSG decreased from 767.0 nm to 332.7 nm after ultrasound treatment, presumed to be caused by the degradation of FSG after ultrasound [47]. Similar results were reported for ultrasonication-treated Persian gum and gum tragacanth [48]. The particle size of both (ZNP-FSG)U and (ZNP-FSG)HU was the lowest among all complexes, and there was no significant difference between them. We found out that if the complex eventually requires ultrasonic treatment, there was no difference in particle whether the intermediate process was homogenized or not. There was no significant difference in particle size of (ZNP-FSG_U_)H and (ZNP-FSG)H, revealing the ultrasound on FSG playing no effect on the particle size of the complex. However, the pre-ultrasonication on zein significantly decreased the particle size of the complex (ZNP_U_-FSG)H ]. Since the particle size of (ZNP-FSG)HU was lower than that of (ZNP-FSG)H, it can be concluded that the extra ultrasonication is effective to further decrease the particle size of the homogenized complex. In conclusion, ultrasonic treatment of different ingredients has a significant effect on the particle size of the complex prepared. However, homogenization on complexes does not have significant effects on the particle size.

PDI is usually used to represent the particle size distribution of dispersion [37]. As shown in Figure 6b, the PDI values of FSG and FSG_U_ are significantly higher than other samples, indicating a broad particle size distribution [29]. The PDI of (ZNP-FSG_U_)H, (ZNP-FSG)HU and (ZNP-FSG)U were higher than that of other complexes, which were all around 0.30. The PDI of (ZNP_U_-FSG)H was the lowest (0.21), showing the uniformity of particle size distribution was increased due to the pre-ultrasonication on zein. The particle size distribution of various ingredients or ZNP -FSG complexes subjected to various ultrasound and homogenization treatments are showing in Appendix A. As is known to all, PDI is not the only reason that describes emulsion stability. We should continue to analyze together with other properties of the complexes. 

#### 3.1.6. ζ-Potential

Figure 6c also shows the ζ-potential of ZNP-FSG complexes subjected to various ultrasound and homogenization processes. The stability of complexes could be studied by measuring the electrostatic stability of the colloidal particles [49]. Particles with ζ-potentials more positive than +30 mV or more negative than −30 mV are normally considered stable [50]. In the absence of FSG, the ζ-potential of the ZNP suspension was around +35 mV, which was due to the fact that the pH was below its isoelectric point (IP) and, therefore, ZNP had a net positive charge. The ζ-potential value of ZNP was increased from +35.2 mV to +45.9 mV after ultrasound pretreatment, suggesting that pretreatment on zein could improve the stability of ZNP. This was consistent with previous results that the ZNP system was stable below pH 6.0 [16]. In fact, ZNP remained stable for several days at 4 °C. The ζ-potential value of FSG was increased from −49.7 mV to −47.1 mV after ultrasound treatment, indicating that ultrasonic treatment slightly lowered the stability of FSG. According to Figure 5, the ζ-potential of complexes were negative, which is because the negatively charged FSG molecules interacted with the ZNP through electrostatic force and aggregates on the surface of positively charged ZNP, resulting in charge reversal. Similar results were found in the study on complexes of WPI gel-chitosan and zein-carboxymethyl dextrin [28,51].

The ζ-potential values of ZNP-FSG complexes subjected to various ultrasound and homogenization treatments were obviously different from each other, suggesting that the pretreatment and operating process significantly affected the stability of the complexes. The absolute zeta potentials of complexes ranking by increasing order is (ZNP-FSG)U ≈ (ZNP-FSG)HU < (ZNP-FSG_U_)H < (ZNP-FSG)H < (ZNP_U_-FSG)H. First, the absolute zeta potential of (ZNP-FSG)U was equal to that of (ZNP-FSG)HU and was the smallest among all. Thus, it can be seen that homogenization of the complex prior to ultrasonic treatment had no effect on their stability. Second, the absolute ζ-potential of (ZNP-FSG)HU was lower than that of (ZNP-FSG)H, indicating the stability of the complex was significantly decreased by ultrasonic treatment after homogenization, regardless of the ingredients of ultrasound. Based on the above results, we concluded that in order to obtain ZNP-FSG complexes with better stability, ultrasound and homogenization cannot be superimposed on the complex. Maybe the mechanical force and cavitation induced by ultrasound increase the energy and entropy of ZNP-FSG system and break the balance made by homogenization, even the particle size or type of interaction force may be affected as well [14]. Ultrasonic treatment has the capability of decreasing the size of protein [52] and reduces the viscosity of polysaccharides by degrading chains of them [48]. The ζ-potential of (ZNP-FSG_U_)H reduced in comparison with ZNP, which was ascribed to the decrease of amounts and distributions of charges on the ZNP surface because of the depolymerizing of FSG during the ultrasonic process [30]. 

However, when applying ultrasonic pretreatment on a single ingredient before homogenization, the complex had better stability. Sample (ZNP_U_-FSG)H showed the maximum absolute ζ-potential value and the best stability, and indicated its better stability than (ZNP-FSG)H or (ZNP-FSG_U_)H. We found that the selection of ultrasonic ingredients significantly affected the stability of the complex. When the ultrasonic pretreatment ingredient was protein, the ultrasonic had a positive effect on the stability of the complex, whereas if the ultrasonic pretreatment ingredient was FSG, a negative effect on the stability of the complex was observed. Some possible reasons for the results were speculated from different mechanisms of ultrasonic treatment on different ingredients. 

In addition, we tried to discuss the relationship between particle size and the potential of the ZNP-FSG complex. There was no direct correlation between the stability and particle size of the complex nor an obvious correlation between ζ-potential and particle size changes of the ZNP-FSG complex. Although ultrasound is widely used in the pretreatment of various samples in preparing a Pickering emulsion for emulsification and homogenization, its effect on homogeneity was revealed for the first time. Above all, the selection of ingredients for treatment and combination of process technology is very important for obtaining a good effect on the stability of protein–polysaccharides complex during the emulsification and homogenization.

### 3.2. Effects of Ultrasound Process on the Emulsion of ZNP-FSG Complexes

#### 3.2.1. Storage Stability of Emulsions

Effects of ultrasonic pretreatment on the physicochemical changes of ZNP-FSG complex and the characteristics of subsequent oil-in-water emulsions were investigated. The creaming index is the ratio of the height of the serum phase to the total height of the emulsion. The creaming index of the emulsions stored at 4 °C for 14 days is shown in Table 2. The creaming index was used as a measure of the stability of Pickering emulsions. The occurrence of the phase separation phenomenon proves that the emulsion has poor storage stability [53]. The appearance of the emulsions prepared with ZNP showed slight creaming as early as 30 min after the emulsions were prepared. The creaming index value of ZNP-E exhibited the highest of 25% after 30 min and then increased to 30%, indicating that ZNP-E possessed the worst stability among all the emulsions. The creaming index of (ZNP-FSG_U_)-E, (ZNP-FSG)HU-E, and (ZNP-FSG)U-E in turn increase. In other words, their stability decreases in turn. However, (ZNP-FSG)-E and (ZNP_U_-FSG)-E remained physically stable for 14 days at 4 °C. This was consistent with the result that their corresponding complexes had the highest EAI and ESI values. 

#### 3.2.2. Freeze-Thaw Stability of Emulsions

The emulsions might undergo the freeze-thaw process. Many emulsions were extremely unstable after three cycles of freeze-thaw treatment, some phenomena such as creaming, flocculation, and coalescence occurred [54]. The freeze-thaw stability of all emulsions stabilized by complexes subjected to various ultrasound and homogenization treatments were evaluated by the creaming index and oiling off and are shown in Figure 7. 

It could be seen from Figure 7 that both the oiling off and creaming index of emulsions increased with the number of freeze-thaw treatment cycles. Obviously, ZNP-E had the highest oiling off and creaming index. Compared with ZNP-E, the oiling off and creaming index of ZNP-FSG complexes subjected to various ultrasound and homogenization treatments all decreased sharply, indicating that the addition of FSG improved the freeze-thaw stability of the emulsion. However, different treatments play important roles in the process. As expected, the oiling off and the creaming index of (ZNP-FSG)U-E and (ZNP-FSG)HU-E were higher than other emulsions except for ZNP-E. Surprisingly, (ZNP-FSG_U_)H-E showed the lowest oiling off, followed by (ZNP_U_-FSG)H-E, and then the (ZNP-FSG)H-E, the sequence of which was different from the order of the creaming index. The creaming index of (ZNP_U_-FSG)-E was the lowest after three freeze-thaw cycles, followed by (ZNP-FSG_U_)H-E and (ZNP-FSG)H-E. 

After the emulsion is frozen, the droplets can undergo coalescence because of the oil phase and water phase crystallization-induced destruction of the emulsions interface film [55]. Then, the addition of FSG could increase the thickness of the interfacial film and provide enough repulsion to prevent droplet aggregation [56]. The improvement of the freeze-thaw stability of the emulsion is closely related to the emulsification and surface hydrophobicity of the ZNP-FSG complexes [57]. As discussed above, ZNP_U_-FSG)H exhibited superior emulsifying properties mainly attributed to the great ability of decreasing the interfacial tension. Thus, a decrease in freeze-thawing stability of (ZNP-FSG)U-E and (ZNP-FSG)HU-E would be directly associated with a combination of reduced emulsification ability and surface hydrophobicity of complexes. Next, the structure of emulsions was further studied.

#### 3.2.3. Emulsion Droplets Size Distribution 

The appearance and droplets size of emulsions (micron scale) stabilized by complexes subjected to various ultrasound and homogenization treatments were presented in Figure 8. The Pickering emulsion of ZNP-FSG can be infinitely diluted in water, indicating they were oil/water emulsions. As shown in Figure 8, all the emulsions exhibited unimodal distributions. It can be observed that emulsion stabilized by individual ZNP tended to separate into two layers. The droplet size of the emulsion decreased from 60.03 ± 5.72 μm to 48.43 ± 3.34 μm with the addition of FSG, and no serum phase was observed. The droplets size of (ZNP-FSG)HU-E and (ZNP-FSG)U-E were 60.03 ± 1.35 μm and 59.97 ± 3.16 μm, respectively. However, the emulsions broke after 30 min at room temperature.

Emulsion droplets size is one of the key parameters to characterize the properties of the emulsion. The droplets size distribution of emulsion depends on the breakup and coalescence of droplets. The high shear force (ultrasound and homogenization) can cause breakup of droplets, and the droplets coalescence may be attributed to surfactants [35]. The presence of FSG decreased the interfacial wettability and promoted complexes to absorb irreversibly on the oil/water interface, which is beneficial to the stability of the emulsion. However, the changing trend of particle size of emulsion was similar to that of complexes, (ZNP_U_-FSG)H-E obtained the lowest droplets size of emulsions (46.9 ± 0.30 μm) because of the small particle size of the complex. According to the results of storage stability and freeze-thaw stability above, (ZNP_U_-FSG)H-E showed relatively good storage stability and freeze-thaw stability compared with other groups of emulsions.

#### 3.2.4. Microstructure of Emulsions

Figure 9 shows the optical microscope photographs and CLSM of Pickering emulsions prepared by complexes subjected to various ultrasound and homogenization treatments. The droplets size distribution is consistent with the result of the droplets microstructure of the emulsion. The droplets of emulsion stabilized by ZNP alone has large droplets size and irregular shape. The creaming index of ZNP-E reached 25% in 30 min and eventually reached the maximum of 30% in 4 days. Pickering emulsions stabilized by ZNP-FSG had no aqueous phase, manifesting that the addition of FSG increased the stability of Pickering emulsions. (ZNP_U_-FSG)H-E showed significantly more droplets and more regular shapes compared to other emulsions, which is consistent with the observations for emulsions stabilized by pea protein with ultrasound [52]. Compared with (ZNP_U_-FSG)H-E, the droplets size of (ZNP-FSG_U_)H-E showed obviously thicker walls and irregular shapes. In the course of observation, we found that the droplet size of (ZNP-FSG)U-E and (ZNP-FSG)HU-E increased gradually over time, and the phase separation phenomenon was observed in 30 min, suggesting their stability were poor. In addition, the presence of oil droplets can be observed clearly on the emulsion droplet surface, which could be explained by two reasons. Firstly, ultrasound increased the frequency of contact collisions between droplets, leading to droplet coalescence. Secondly, over-processing of complexes was caused by ultrasonic and high-speed homogenization.

Figure 9b shows CLSM images of Pickering emulsions after 24 h storage at 4 °C. Corn oil was dyed with Nile red. All the above results showed that homogenization had no significant effect on the complexes and emulsion, thus (ZNP-FSG)U-E was not shown in the CLSM diagram. (ZNP_U_-FSG)-E can prevent collisions and coalescence of droplets and contained a fairly uniform dispersion of spherical oil droplets. The bigger oil droplet size was observed in (ZNP-FSG_U_)-E and (ZNP-FSG)HU-E. The oil droplet size variation observed by CLSM was in agreement with the storage stability.

## 4. Conclusions

In summary, different ultrasonication and homogenization treatments combination on zein and FSG can significantly influence the properties of the complexes and the emulsion formed by the complexes. Ultrasonic treatment affected the interaction between ZNP and FSG. The complex formed by ZNP_U_ and FSG has the highest absolute zeta potential value, better emulsification properties (higher EAI and ESI), and the highest surface hydrophobicity among all, which indicates that the complex was stable and can be used as a good Pickering emulsions stabilizer among those complexes subjected to various ultrasound and homogenization treatments. However, the properties of (ZNP-FSG)U and (ZNP-FSG)HU are very different from those of (ZNP_U_-FSG)H. (ZNP_U_-FSG)H-E with good storage stability and freeze-thaw stability can be observed in the microscope with a clear, regular, and uniform spherical structure. In conclusion, these results indicate that the timing of using ultrasound in the preparation of complexes and emulsions is very important. These results provide new ideas and a theoretical basis for improving the properties of protein–polysaccharide complexes and emulsions.

## Figures and Tables

**Figure 1 foods-10-01990-f001:**
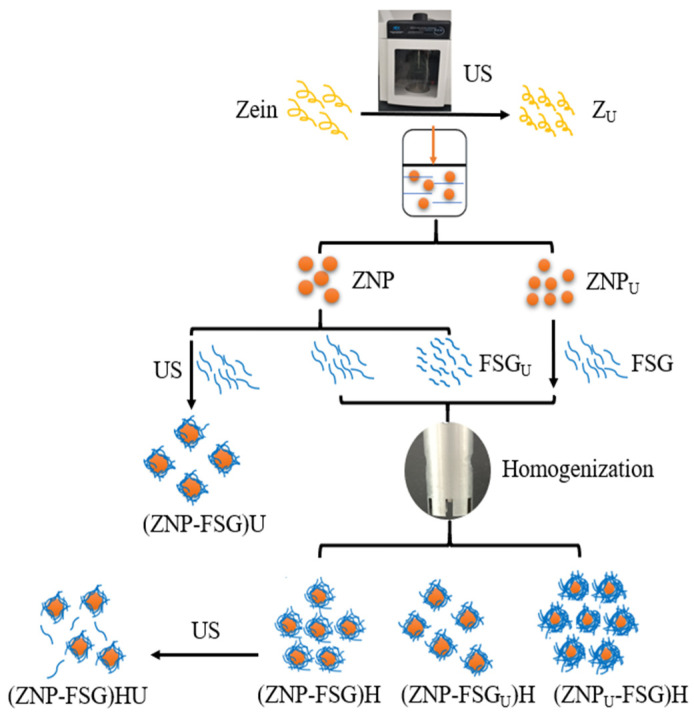
Overall schematic diagram of 11 various preparation routes, such as ultrasonicated zein (Z_U_), zein nanoparticles (ZNP), ultrasonicated zein nanoparticles (ZNP_U_), flaxseed gum (FSG), ultrasonicated flaxseed gum (FSG_U_), zein nanoparticles-flaxseed gum complexes by homogenization (ZNP-FSG)H, zein nanoparticles-flaxseed gum complexes by ultrasonication (ZNP-FSG)U, ultrasonicated zein nanoparticles and flaxseed gum complex by homogenization (ZNP_U_-FSG)H, zein nanoparticles and ultrasonicated flaxseed gum complex by homogenization (ZNP-FSG_U_)H, ZNP-FSG by homogenization and ultrasonication (ZNP-FSG)HU.

**Figure 2 foods-10-01990-f002:**
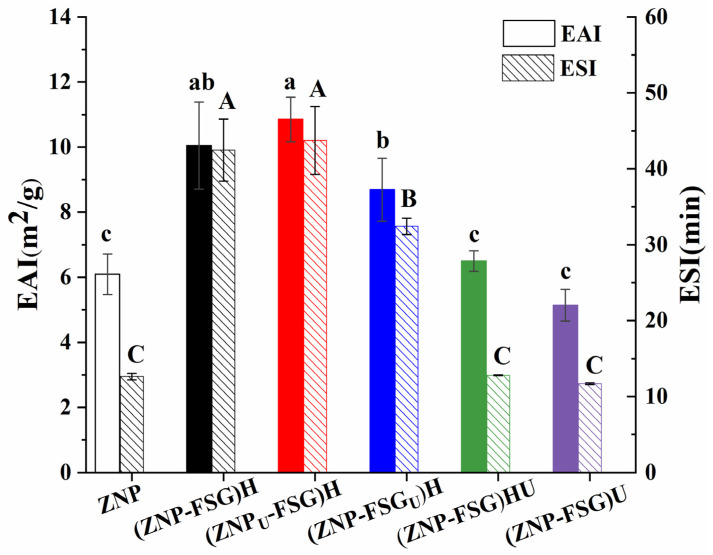
Emulsifying activity index (EAI) and emulsifying stability index (ESI) of emulsions stabilized by ZNP-FSG complexes subjected to various ultrasound and homogenization treatments.

**Figure 3 foods-10-01990-f003:**
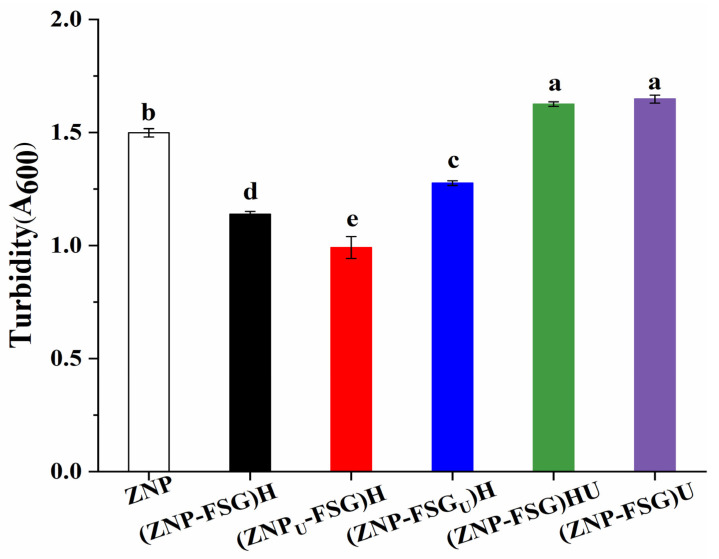
Turbidity of ZNP-FSG complexes subjected to various ultrasound and homogenization treatments.

**Figure 4 foods-10-01990-f004:**
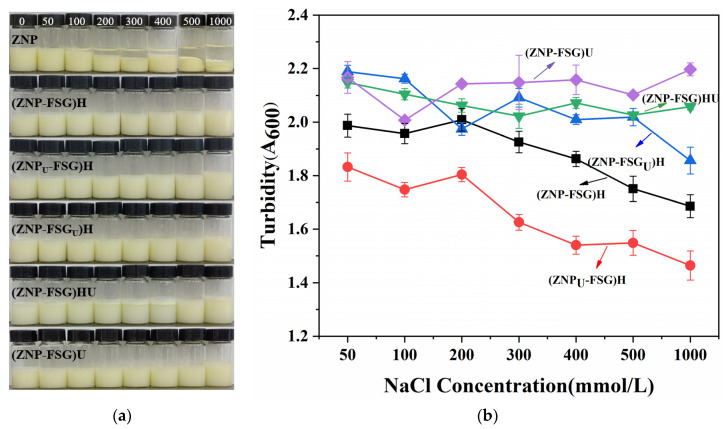
Visual appearance (**a**) and turbidity (**b**) of various ZNP-FSG complexes prepared in different NaCl concentration, i.e., black for (ZNP-FSG)H, red for (ZNP_U_-FSG)H, blue for (ZNP-FSG_U_)H, green for (ZNP-FSG)HU, and purple for (ZNP-FSG)U.

**Figure 5 foods-10-01990-f005:**
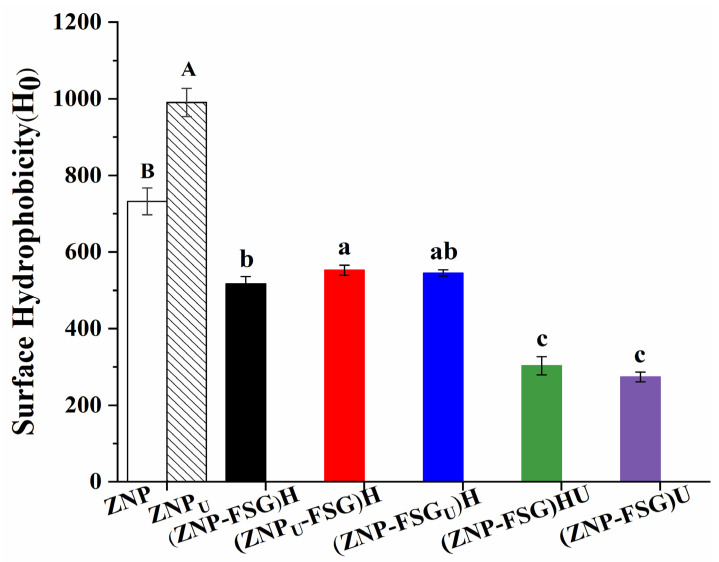
Surface Hydrophobicity (H_0_) of ZNP-FSG complexes subjected to various ultrasound and homogenization treatments.

**Figure 6 foods-10-01990-f006:**
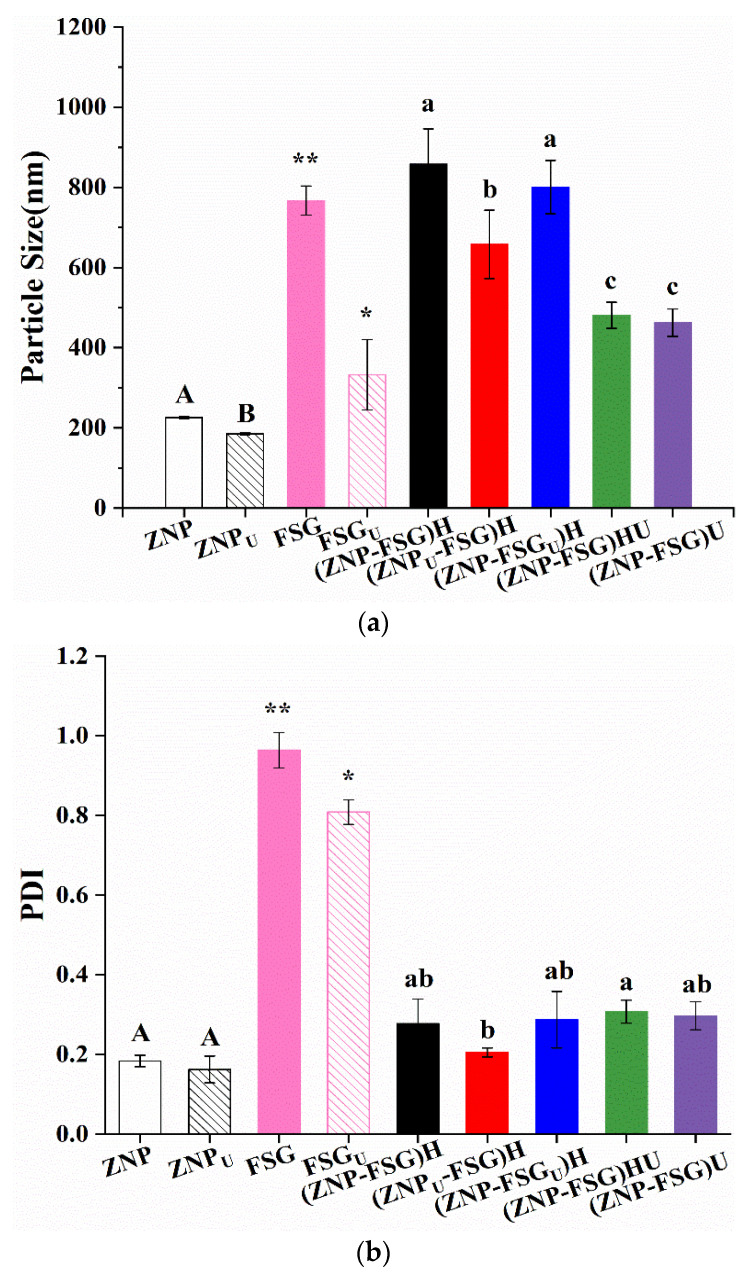
Particle size (**a**), PDI (**b**), and ζ-potential (**c**) of ZNP-FSG complexes subjected to various ultrasound and homogenization treatments. Different lowercase letters (a, b), different capital letters (A, B) and asterisk (*, **) indicated significant difference (*p* < 0.05).

**Figure 7 foods-10-01990-f007:**
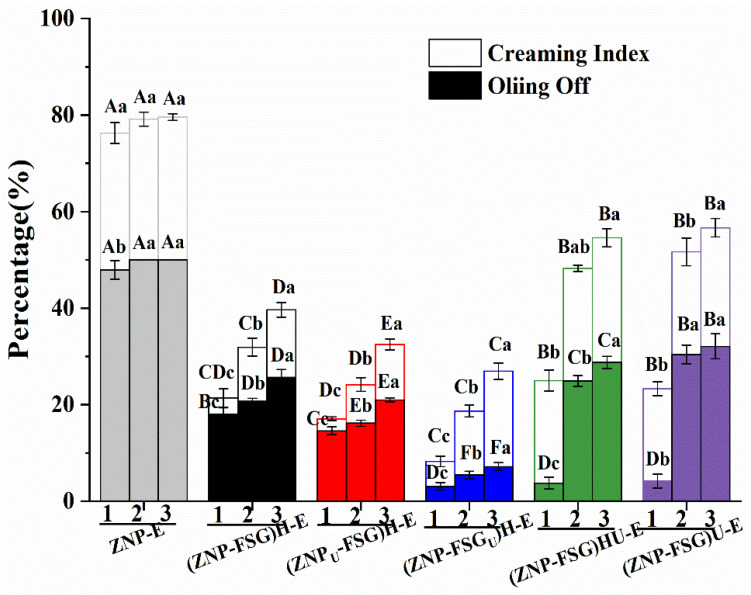
The oiling off and creaming index of the Pickering emulsions stabilized by ZNP-FSG complexes subjected to various ultrasound and homogenization treatments. 1, 2, 3 represent freeze-thaw cycles, respectively.

**Figure 8 foods-10-01990-f008:**
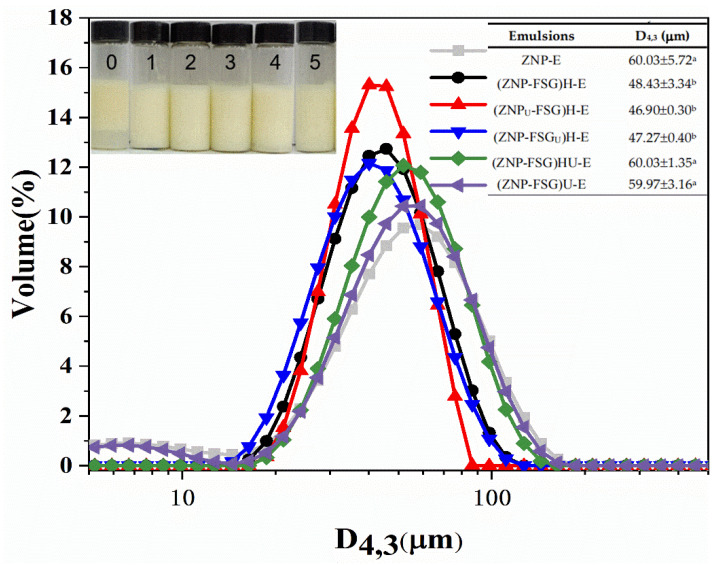
The particle size distribution of emulsions stabilized by ZNP-FSG complexes subjected to various ultrasound and homogenization treatments. Insets are photographs of fresh emulsions (0. ZNP-E, 1. (ZNP-FSG)H-E, 2. (ZNP_U_-FSG)H-E, 3. (ZNP-FSG_U_)H-E, 4. (ZNP-FSG)HU-E, 5. (ZNP-FSG)U-E). The emulsion droplets sizes are shown in the table.

**Figure 9 foods-10-01990-f009:**
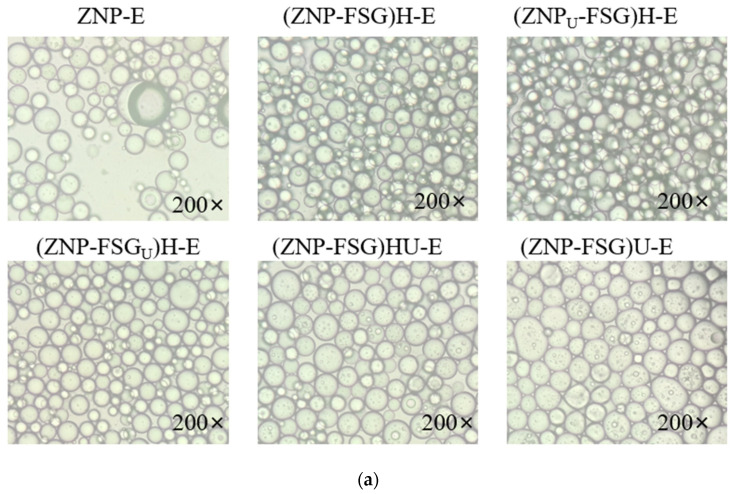
Optical micrographics (**a**) and Confocal laser scanning microscopy (CLSM) (**b**) of emulsions stabilized by ZNP and FSG complexes subjected to various ultrasound and homogenization treatments.

**Table 1 foods-10-01990-t001:** The process of complex particles formation.

Sample No.	Complexes	Materials		Process	
		Protein	Polysaccharides	Homogenizing (rpm/3 min)	Ultrasound (W·cm^−2^)
1	ZNP	ZNP	-	-	-
2	ZNP_U_	ZNP_U_	-	-	52.95
3	FSG	-	FSG	-	-
4	FSG_U_	-	FSG_U_	-	52.95
5	(ZNP-FSG)H	ZNP	FSG	15,000	-
6	(ZNP_U_-FSG)H	ZNP_U_	FSG	15,000	52.95
7	(ZNP-FSG_U_)H	ZNP	FSG_U_	15,000	52.95
8	(ZNP-FSG)HU	ZNP	FSG	15,000	52.95
9	(ZNP-FSG)U	ZNP	FSG	-	52.95

“-” in Materials indicates without the addition of a certain ingredient; “-” in Process indicates the sample without treatment by ultrasonication or homogenization.

**Table 2 foods-10-01990-t002:** The creaming index (%) of emulsions stabilized by ZNP-FSG complexes subjected to various ultrasound and homogenization treatments.

Creaming Index (%)	Time								
	0.5 h	2 h	1 d	2 d	3 d	4 d	7 d	10 d	14 d
ZNP-E	25	25	25	27.5	30	30	30	30	30
(ZNP-FSG)H-E	0	0	0	0	0	0	0	0	0
(ZNP_U_-FSG)H-E	0	0	0	0	0	0	0	0	0
(ZNP-FSG_U_)H-E	7.5	7.5	8	10	10	10	25	25	25
(ZNP-FSG)HU-E	17.5	22.5	25	25	25	25	25	25	25
(ZNP-FSG)U-E	17.5	25	25	25	25	25	25	25	25

## Data Availability

The datasets generated for this study are available on request to the corresponding author.

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
