# Peer review of "The Role of Ultrasound in the Preparation of Zein Nanoparticles/Flaxseed Gum Complexes for the Stabilization of Pickering Emulsion"

_foods, 2021, doi:10.3390/foods10091990_

Round 1

Reviewer 1 Report

The AUthors wanted to present a Pickering type emulsion characterization. Zein NPs were used as a stabilizer. First what I like in this manuscript, that you used different techniques for the NP as well as emulsion characterization techniques, such as DLS and microscopy, but... There is a lot to do before your results description will be ready for publishing.

First of all, how can you prove that Zein NP stabilizes the system - please highlight it in the text. How you can confirm the Pickering type emulsion? Then see detailed comments listed below, as well as recommended manuscripts that could help you to enhance the quality of the description and quality of the results of the manuscript. 

Introduction

  • shortly define Pickering emulsion
  • extend introduction of  zein/flaxseed gum system and present the other studies 
  • how the pH was adjusted?
  • l. 87 stirring or ultrasound treatment?
  • p. 2.2.3. please change the description - I have no idea what was ultrasound treated.
  • p. 2.3.1., 2.3.2. - change it for table
  • p. 2.5.1. describe more details like scattering angle,  final pH, how many times it was repeated
  • p.2.5.2. rewrite first sentence
  • p.2.5.3. please express in more detail how the hydrophobicity can be expressed by fluorescence? Usually, the contact angle is measured.
  • p.2.7. what you measured here. How did you determine RI? 

Please follow by this paper how the description of the experiment should look like:

Int. J. Mol. Sci. 202122(2), 887; https://doi.org/10.3390/ijms22020887

Molecules 202126(10), 2931; https://doi.org/10.3390/molecules26102931

  1. 29 - what do you mean by traditional emulsion?

Please update the references due to Journal style.

Please extend the introduction of other studies.

Results and discussion:

  • please uniform the colors in all figures for each sample
  • Salt ionic stability  - wherein the materials and methods it was described?
  • The discussion and the presented results of particle size studies should be totally rewritten.
  • what you presented in the table: z-ave, peak maximum, or what?
  • what was the PDI of the samples

Reviewer 2 Report

The material presented in the article is correct. The goal is clearly defined and the methodology is correctly described. The research methods were appropriately selected to document the properties of the tested emulsions. The results were presented clearly, correctly interpreted. The conclusions result from the presented research.
The text requires an editorial correction, especially in the case of missing or additional spaces.
Detailed comments:
Figure 2
Homogeneous groups should be marked with different letters for the EAI parameter, e.g. a, b, c), different for the ESI parameter (e.g. A, B, C…).
space is needed
Line 156. [22] .30 mL
Line 212 and FSG [22]
Line 225 "[26] studied" -
Line 230 bridge [28]
Line 232 FSG [25]
Line 240 and 245: effects [29]
Line 256 properties [28]
Line 268 et.al [31]
Line 278 al. [32].
Line 371 stability [42]
Line 404 complexes [46]
Line 452 ultrasound [41]
Space is not needed:
 line 239 and b). In
line 244: of the
line 375, The dot and not a comma: increase, In
Verify spaces
Line 440 0.ZNP-E, 1. (ZNP-FSG) HE, 2. (ZNP U -FSG) HE, 3. (ZNP-FSG U) HE, 4. (ZNP-FSG) HU-E, 5 . (ZNP-FSG) UE

Reviewer 3 Report

The manuscript describes the preparation of zein nanoparticles-flaxseed gum (ZNP-FLG) complexes according to different procedures, which imply either the ultrasonication or the homogenization of the mixed solution of the two partners without any previous treatment or after ultrasonic treatment of the solutions of only one or both ingredients. The aim is to devise the preparation route of ZNP-FLG complexes best suited for the stabilization of O/W Pickering emulsions. 

Both the ZNP-FLG complexes and the final emulsions were characterized. The former under the respect of size, zeta potential, stability at different ionic strength, hydrophobicity and the latter by considering the size and shape of the oil droplets and the emulsion stability.

1) In my opinion there is missing information in the experimental procedure, in detail:

It would be useful to state soon that HCl and NaOH 0.1M solutions were used to adjust the pH  of both deionized water and of the various prepared systems to 4 and that the pH was measured by a pHmeter, instead of mentioning it only in section 2.5.1

In the case of surface hydrophobicity determination by means of a fluorescent probe, the wavelength of the maximum in the fluorescence spectrum should be reported, as in ref 24, where it is mentioned that the fluorescence was measured at 468 nm.

Concerning the hydrophobicity determination by means of an anionic probe also the effect of electrostatic interaction  must be considered. see Comparison of Protein Surface Hydrophobicity Measured at Various pH Values Using Three Different Fluorescent Probes Nooshin Alizadeh-Pasdar and Eunice C. Y. Li-Chan Journal of Agricultural and Food Chemistry 2000 48 (2), 328-334 DOI: 10.1021/jf990393p

Neither it can be safely stated (line 260) that surface hydrophobicity “indicates” the number of hydrophobic groups, rather “is related to   

2) As an overall the work is difficult to read because of the writing a bit confuse and with various not appropriate terms and many grammar mistakes, therefore a thorough, accurate language check is required. A few suggestions are listed below:

-starting from the title, where it is not clear that ultrasound are used in the preparation of the ZNP-FLG complexes. “The role of ultrasound in the preparation of zein nanoparticles/flaxseed gum complexes for the stabilization of Pickering emulsion” would be clearer

-Abstract as well does not reflect faithfully the content of the work

line 9: “to prepare” is more appropriate than "to synthesize”

line 10: “and” rather than “which”

line 12: formation of zein nanoparticles/flaxseed gum complexes

the preparation route of the ZNP should be mentioned. Moreover, it should be clearer that the effect on the characteristic features of ZNP-FLG complexes of the employ of ultrasounds during preparation was investigated.

-line 123 The emulsion were prepared by mixing …

-lines 129-131 were analyzed by dynamic light scattering using a Zetasizer …

-line 132 detection

-line 135 Turbidity was determined according to Hosseini [20]

-line 145 series

-line 148 of the diluted sample 

-lines 155-156 was carried out according to the method of Li et al. [22], slightly changed.

-line 158 SDS explain the acronym, even if is obvious

-line 159 Afterwards the fresh…

-line 163 “optical path” instead of “thickness”

-line 164 t as small letter because it refers to time

-line 167 The method of Ahmed et al. [10] with slight modifications was used to assess the storage stability of Pickering emulsions

-line 175 milliliters?

-lines 223-224 are very obscure and the whole Results and Discussion section could be improved under the respect of clarity

-ref 14 missing authors’ surnames 

Round 2

Reviewer 1 Report

The Authors strongly improved manuscript quality. There are still my concerns about the DLS result presentations.  Detailed comments are listed below.

  1. 11 various preparation routes... - add such as...

Uniform colors i.e. green for sample 1, blue for sample 2, add a description in fig. 4 for the curves.

Description of PDI is weak please add some comments and references. What is more, give obtained values range - it is not enough what you presented on the graph (PDI is not higher than 1). Just keep in mind that PDI is not the only reason which describes emulsion stability - even EM with high PDI can be stable.

Section particle size - what do you present here z-ave or peak maximum? As previously mentioned particle size distribution graph should be added i.e. supplementary materials.

Fig. 9 adds a more visible scale bar.
